# Working in the Times of COVID-19. Psychological Impact of the Pandemic in Frontline Workers in Spain

**DOI:** 10.3390/ijerph17218149

**Published:** 2020-11-04

**Authors:** Rocío Rodríguez-Rey, Helena Garrido-Hernansaiz, Nereida Bueno-Guerra

**Affiliations:** 1School of Human and Social Sciences, Department of Psychology, Comillas Pontifical University, 28015 Madrid, Spain; rocio.r.rey@comillas.edu; 2Department of Education and Psychology, Centro Universitario Cardenal Cisneros, 28806 Alcalá de Henares, Spain; helenagarrido42@gmail.com

**Keywords:** COVID-19, pandemic, psychological impact, frontline workers, depression, occupational health, healthcare providers, journalists, grocery store workers, protective service workers, Spain

## Abstract

This study evaluates the psychological impact (PI) of the COVID-19 pandemic in frontline workers in Spain. Participants were 546 workers (296 healthcare workers, 105 media professionals, 89 grocery workers, and 83 protective service workers). They all completed online questionnaires assessing PI, sadness, concerns related to the COVID-19 pandemic, and demographic and work-related variables. All groups but protective services workers showed higher PI levels than the general population. Healthcare and grocery workers were the most affected, with 73.6% and 65.2% of the participants, respectively, showing a severe PI. Women showed a higher PI level. Healthcare workers in the regions with higher COVID-19 incidences reported greater PI levels. The main concerns were being infected by COVID-19 or infecting others. Levels of concern correlated with higher PI levels. The protection equipment was generally reported as insufficient, which correlated with higher PI levels. Professionals reporting to overwork during the crisis (60% mass-media, 38% of healthcare and grocery and 21.7% of protective service) showed higher PI levels. In the healthcare group, taking care of patients with COVID-19 (77%) or of dying patients with COVID-19 (43.9%) was associated with higher PI levels. The perceived social recognition of their work was inversely related to PI. Most of the sample had not received psychological support. We suggest some organizational measures for frontline institutions, such as the periodical monitoring or inclusion of psychologists specialized in crisis-management to prevent negative symptoms and provide timely support.

## 1. Introduction

Spain was a front door country during the first weeks of the COVID-19 pandemic spread in Europe. As of 17 September 2020, more than 600,000 infected cases and more than 30,000 deaths (within a total population of 47 million people [1]) have been reported in this country [2,3]. The situation was especially hard in the period between March and May 2020, when the number of cases and deaths increased rapidly and, therefore, the Spanish Government mandated a national lockdown that started on March 15th. As is being increasingly reported, measures taken to control the COVID-19 pandemic such as the lockdown have caused a severe impact on mental health worldwide. Three studies conducted with the Spanish general population during this period [4,5,6] revealed significant degrees of psychological impact, post-traumatic symptoms, anxiety, depression, and stress. These three studies reported that women, younger participants, students, and people with economic difficulties showed a higher symptomatology. During the aforementioned lockdown, the essential services providers continued working [7,8] and had to deal with the consequences of the pandemic in their job. That is why essential workers are especially vulnerable to the psychological effects of the pandemic. In fact, Rodríguez-Rey et al. [6] and González-Sanguino et al. [4] found that people who worked on-site during the mandated lockdown showed higher distress levels than people who were teleworking.

During the COVID-19 pandemic, frontline workers bumped into a succession of first-hand, first-time seen traumatic images, such as empty main streets, the overcrowding of coffins, and war-like number of critic patients with a great scarcity of personal protection equipment, therefore risking their and their relatives’ lives [9,10]. Understanding how this new scenario could have an impact on the psychological health of frontline workers is relevant to provide accurate post-crisis assistance, as well as to compile data that could be useful to prevent mental disorders in outbreaks or new pandemics. Therefore, the objective of the present study was to explore the psychological impact of the COVID-19 pandemic on frontline workers and to explore its association with multiple work-related variables, related concerns, and sociodemographic characteristics.

Following the Spanish legal disposals [8,9], essential services can be divided into seven areas: health (e.g., health care personnel), provisioning and basic services (e.g., groceries, stores, and supermarkets), information and communication (e.g., journalists and mass-media personnel), security (e.g., army, police, and rescue services), transportation (e.g., fuel stations and public transportation), finances (e.g., banks), and miscellany (e.g., funeral parlors and tobacco shops). The consideration of essential workers encounters an international general agreement with few variations and always highlights the special relevance of frontline workers, namely those who cannot feasibly work from home and therefore are more exposed to contagions. Therefore, in accordance with both Spanish and international criteria, in this study, we considered frontline workers as healthcare workers, protective service workers, cashiers in grocery and general merchandise [11], and journalists working during the Spanish lockdown period.

Healthcare workers normally experience higher burnout, depression, anxiety, and suicide risk than individuals with other occupations (e.g., [12]). During the COVID-19 crisis, healthcare workers are facing extraordinary amounts of pressure, excessive workloads, uncertainty, emotional overburden, and high exposure to the virus [13]. Intensive care units have surpassed their capacities so much that regular hospitals have had to occupy beds previously assigned to different services. Emergency field hospitals have been set up, and physicians with non-emergency specializations have had to move to assist COVID-19 patients [14,15]. Additionally, healthcare workers have frequently lacked sufficient personal protective equipment and have felt inadequately protected against the virus [13,16]. Therefore, burnout, depression, and anxiety levels have dramatically increased. According to a meta-analysis that included 13 studies carried out in Asian countries, anxiety and depression are currently present in more than 20% of the healthcare workers [17]. In Spain, the situation is similar, with a very high psychological impact reported in different studies [18,19,20]. Healthcare workers at respiratory medicine services, tertiary hospitals, regions with the highest incidence of COVID-19 [20], as well as those with a presence of COVID-19 symptoms and virus contact history [19], have shown the highest distress levels. Some other relevant variables associated with higher distress are the lack of previous crisis-management experience [21,22], younger age [20,23], female gender [17,24], the prevalence of physical symptoms [25], and factors that increase the risk of becoming infected such as being in direct contact with patients and shortages of personal protection [26].

As for grocery workers, food purchasing patterns have changed dramatically during the pandemic, turning from regular visits to shop both perishable and non-perishable products to sporadic hoarding behaviors such as non-perishable stockpiling or panic buying [27]. Thus, “food supply chains needed to adjust rapidly to demand-side shocks, as well as plan for any supply-side disruptions due to potential labor shortages and disruptions to transportation” [27], (p. 171). Moreover, the shopping experience enormously changed during the lockdown period, with the occurrence of long queues, limited capacity restrictions, maintained distances, limited shopping times, and the mandatory use of masks and sometimes gloves [28]. Furthermore, due to their lack of information on how to manage sanitary crises, to reduce their risk of infection and their level of uncertainty, grocery workers have faced an unprecedented scenario during the pandemic and consequently experienced high distress [29].

Regarding protective service workers such as police, militaries, and public safety workers, they frequently face an enormous diversity of physical and psychological hazards [30] that may cause severe negative psychological consequences. Indeed, a recent issue concluded that mental disorders are the second most common reason for medical discharge in some uniformed services [31]. Concretely, among police workers, patrolling is the main source of anxiety [32], but one-time severe crises may cause a myriad of psychological disorders. For instance, after the 11-M terrorist attack in Madrid in 2004, almost 4% of police workers involved in the rescue reported symptoms consistent with psychiatric disorders [33]. During the COVID-19 pandemic, public safety and security workers in the occupational sectors have been the most affected by infections [34]. Notably, as their work necessitates dealing with emotionally demanding situations, this sector scores high in resilience (probably due to cumulative emergency exposure [31]). However, presenteeism (attending work when lacking health) is a common thread due to their usual reluctance to admit distress [35].

Finally, media professionals rarely appear as frontline workers in the literature except when in conflicts and war periods (e.g., [36]). In those cases, it has been largely reported that conflict journalists suffer more from psychological distress than local journalists (e.g., [37])—they end up “depleted and broken, their personal lives often in shambles, their sleep plagued by images of death” [38], (p. 9). Concretely, it has been found that the amount of previous crisis-related assignments, rather than a protective factor, become a risk factor for higher levels of post-traumatic symptoms [39]. During the whole COVID-19 pandemic, news on television and radio has continued broadcasting, and newspapers have continued being printed and published online. Thus, mass-media professionals, including not only visible faces (i.e., correspondents, presenters, and journalists) but also technicians (e.g., camera and sound technicians), have played a critical role given that they must clearly, sensitively, and regularly provide accurate and constantly updated information. Indeed, media professionals have acquired some societal responsibility since this information could contribute to reducing people’s uncertainty and fear [40] and fighting misinformation [41]. Moreover, the work of media professionals during the pandemic has entailed being constantly updated and coping with a constant flow of data that implies rapid fact-checking through very different simultaneous and varying sources to keep the population constantly informed, therefore being unable to disconnect digitally or to balance family and work. To our knowledge, there has been no study covering how the pandemic has affected the psychological health of media professionals. However, this overwhelming information management is relevant given that information overload has been found to be associated with poorer mental health outcomes in the general population [6].

Hence, once all the peculiarities of working during the COVID-19 pandemics in each frontline group of workers are stated, we present a study in which we aimed to explore their psychological symptomatic response to working during the COVID-19 pandemic and the potential demographic and work-related factors that may be associated with their symptoms. Additionally, such psychological symptomatic responses were compared across the four groups and with the data obtained from the Spanish general population in the study by Rodríguez-Rey et al. [6]. By presenting this knowledge, we hope to be more capable of taking care of those who are taking care of us.

## 2. Materials and Methods

### 2.1. Participants

A total of 546 frontline professionals completed the study. Of those, 49.3% were healthcare workers (*N* = 269), 19.2% were media professionals (*N* = 105), 16.3% were grocery workers (*N* = 89), and 15.2% were protective service workers (*N* = 83). All the participants were currently working or had worked at any time during the first wave of the COVID-19 sanitary crisis in Spain (March–May 2020). Thus, professionals who had not worked during the crisis were excluded from the study. Each subsample’s sociodemographic data are included in Table 1.

### 2.2. Materials


Demographic questions: Participants provided information regarding their gender, age, Spanish province of residence, marital status, number of children or dependent relatives, and educational level.Work-related information: Some questions were shared among all the professional groups, whereas others varied. All the participants were asked about their years of experience in the current position, their previous experience working in critical situations (whether they had worked in a major critical situation before, and, if so, in how many crises), the average number of hours they had worked the week before, whether they had to overwork during the pandemic (and, if so, how many extra hours during the week before), the number of days since their last day off, the degree to which they perceived that they were working as a team during the crisis, and the availability of personal protection equipment in their workplace (those who telework did not have to answer such question). Healthcare workers were also asked to specify their position (e.g., physician, nurse, or nursing assistant), their current work situation (currently working, isolated due to COVID-19 infection, or on leave), the unit where they usually worked (e.g., Intensive Care Unit emergency room, or primary care), the unit where they were working during the COVID-19 crisis, and their proximity to COVID-19 patients (whether they were assisting diagnosed, severe, or dying patients). Media professionals were asked about their working role (e.g., press conferences, interviews, covering breaking news, or technical assistance), how much time they spent in direct contact with other people at work, the media they were working for during the crisis (e.g., written press, digital press, television, radio, or news agency), their media’s audience impact (local, regional, or national), and whether they were teleworking or working on site. Grocery workers were asked about their working role (e.g., cashiers, stock replenisher, customer service, security, or human resources), how much time they spent in direct contact with other people at work during the pandemic and the size of the store (e.g., corner shop, supermarket, or hypermarket). Finally, protective security workers were asked about the force unit they belonged to (civil guard, local police, national police, or army) and their working role during the crisis (e.g., patrolling, providing public information, disinfection, logistics, product supplying, or border or road control).Impact of Event Scale-Revised (IES-R; [42,43]). The IES-R is a self-administered 22-item questionnaire designed to measure the magnitude of a symptomatic response to a specific traumatic life event during the past week. The IES-R measures the three main symptoms of post-traumatic stress disorder (PTSD): avoidance, intrusion, and hyperarousal. The response format was a 5-point scale ranging from 0 (not at all or hardly ever) to 4 (extremely). The total score was obtained by adding the item responses. The IES-R has been validated in the Spanish general population [44] and in Spanish patients with cancer [45], showing adequate psychometric properties. Following the criteria used in the Chinese [46] and the Spanish [6] general populations, the total IES-R score was divided into 0–23 (normal), 24–32 (mild psychological impact), 33–36 (moderate psychological impact), and >37 (severe psychological impact). In the present study, the instructions and the items were adapted to refer to the current COVID-19 sanitary crisis. The internal consistency of the scores was good for both the three subscales (avoidance: α = 0.86; intrusion: α = 0.89; hyperarousal: α = 0.87) and the total scale (α = 0.94).Screening of depression: We asked the participants two questions: (1) Are you depressed? (response options: “Yes,” “I don’t know,” and “No”), which has proven to be an adequate screening question to detect depression with a sensitivity of 88% when the answers “Yes” and “I don’t know” are combined [47], and (2) How sad are you feeling? (response format of 0–10), which has shown adequate sensitivity and specificity to detect emotional distress in previous studies (e.g., [48]).Degree of concern: Participants were asked about the degree to which they were concerned (i.e., not concerned at all, barely concerned, somewhat concerned, rather concerned, or very concerned) about different aspects in relation to their work during the COVID-19 pandemic. Some questions were common for all the groups, while some others were specific for each profession. The full list of assessed concerns is included in the concerns subsection within the results.Perceived social recognition and self-assessment of the importance of their work during the pandemic: Participants were asked whether they felt their work was recognized and whether they thought that their labor contribution was important during the pandemic (i.e., not at all, barely, somewhat, rather, or a lot).Support received by their company/workplace during the COVID-19 crisis: Participants were asked whether they were receiving any specific support to face the current crisis, and, if so, which support (psychological support, financial incentive, or change in labor conditions such as making schedules more flexible).Perceived severity of the COVID-19 crisis: Respondents indicated the perceived severity degree of the current crisis through a Likert scale ranging from 0 to 10.


### 2.3. Procedure

The study was approved by the Ethics Committee of the first and the third authors’ university. The data were collected online through a Google Forms questionnaire specific to each professional group. The four groups included on the study were selected based on the previous crisis-related literature and the international consideration of essential workers, as described in the introduction. The data collection period was between 21 March 2020 (six days after the alarm state was implemented in Spain) and 4 June 2020 (once the 80% of the country had reached Phase 2 in the progressive reopening, which meant that non-essential workers could restart working on-site if telework was not possible). To recruit the sample, we distributed the questionnaires by email and social networks (Facebook, Instagram, Twitter, LinkedIn, and WhatsApp) following a snowball approach. Additionally, we contacted different professional networks and companies and asked them to spread the questionnaires. Specifically, for the healthcare workers group, we contacted the College of Physicians, several public tertiary hospitals located in different Spanish provinces (e.g., Hospital Universitario La Paz, Hospital 12 de Octubre, Hospital Sant Joan de Deu, and Hospital Universitario de Badajoz), and the Alliance of private Spanish healthcare. For the media professionals, we contacted the Spanish Society for Scientific Communication, journals or agencies specialized in science (such as SINC and EFE), the Madrid Press Association (which published the link in their website and distributed it among their members—this encouraged other regional associations, e.g., Catalonia and Aragon, to do the same), university press offices and individual journalists or media workers of Spanish public television (RTVE) or private channels (e.g., Antena3 and La Sexta), radios (e.g., COPE and Cadena SER), and journals (e.g., Público) covering different political orientations. For protective service workers, we contacted the Secretary of State, the National Union of Heads and Directors of Local Police, some Twitter accounts managed by local protective service units, high commands (e.g., sergeant and captain) in military units involved in COVID-19 tasks (e.g., the military emergency unit in charge of disinfection and corpse management) who distributed the study information among their subalterns, and individual workers in the national and local police (e.g., Madrid) and regional civil guards. Finally, for grocery workers, we did a search of supermarkets on a national scale, and then we distributed the study information among restaurant owners who had trade agreements with regional markets and sent emails asking the respective human resources departments to spread the questionnaires. All respondents provided informed consent prior accessing the questionnaires.

### 2.4. Data Analyses

Descriptive statistics were computed for the sociodemographic characteristics of the sample and the study variables consisting of frequencies and percentages for categorical variables and means and standard deviations (SDs) for scale variables. For the IES-R, standardized factorial scores were obtained based on the measurement model of the scale. The Q–Q plots were visually inspected, and skewness and kurtosis values were obtained; all of them indicated normality.

Then, differences in the IES-R mean scores between each group and the general population (using the mean provided by Rodríguez-Rey et al. [6]) were explored via *t*-tests. We also used *t*-tests to look for differences in the means of symptoms when the independent variable was dichotomic (e.g., attending patients with COVID-19 or not), adjusting the test result for non-homogeneous variances if the Levene’s test was significant. For multiple-category variables (e.g., professional group), we used a one-way ANOVA and looked at post-hoc Tukey (for homogeneous variances) or Games–Howell (for non-homogeneous variances) tests in case of a significant *F*-value. We obtained appropriate effect size statistics that adjusted for differences in group sizes—Hedges’ g for *t*-tests and η^2^ for ANOVAs. Bivariate associations between psychological impact and continuous variables (e.g., age) were assessed via Pearson’s correlation coefficient *r*. We used the Spearman’s correlation coefficient *ρ* to test bivariate associations between psychological impact and ordinal variables (e.g., perceived social recognition). All the tests were two-tailed, with a significance level of *p* < 0.05. The statistical analyses were performed using IBM SPSS Statistics for Windows, version 25 (IBM Corp., Armonk, N.Y., USA).

## 3. Results

### 3.1. Psychological Impact and Indicators of Sadness

Table 2 shows the IES-R scores, namely the psychological impact caused by working during the COVID-19 pandemic and the depression indicators. The ANOVAs tests show whether these results varied significantly depending on the professional group to which the participants belonged. Most of the professionals showed severe psychological impact derived from working during the COVID-19 crisis according to the established cut-offs for the IES-R: healthcare workers at 73.6%, grocery workers at 65.2%, and media workers at 48.6%. Interestingly, however, the protective service workers followed a different pattern, with most of the participants showing minimum psychological impact (54.2%). There were significant differences in the levels of psychological impact between groups—the highest mean score was for healthcare and grocery workers, and the lowest score was for protective service workers. Moreover, healthcare workers (*M* = 44.85, *t* = 18.84, and *p* < 0.001), grocery workers (*M* = 42.07, *t* = 7.03, and *p* < 0.001), and media professionals (*M* = 36.20, *t* = 5.65, and *p* < 0.001) showed significantly higher psychological impacts than the general population (*M* = 27.95), while protective service workers showed similar levels (*M* = 25.95, *t* = −1.24, and *p* = 0.22).

There were significant differences in the levels of sadness (range 0–10) between groups. Healthcare workers exhibited the highest scores, followed by grocery workers, media professionals, and, finally, protective service workers. Regarding the question “Are you depressed?” 49.8% of the participants in the healthcare workers group and 56.2% in the grocery workers group answered maybe or yes, while this percentage was 37.1% for media professionals and 18.1% for protective service workers.

Women showed significantly higher psychological impacts than men both in the healthcare worker group (*M*_women_ = 0.41, *SD* = 0.75, *M*_men_ = −0.82, *SD* = 1.02, *t* = −3.99, and *p* < 0.001) and in the grocery worker group (*M*_women_ = 0.35, *SD* = 1, *M*_men_ = −0.67, *SD* = 1.02, *t* = −3.79, and *p* < 0.001). No gender differences were found in media professionals (*t* = −1.87 and *p* = 0.06) or protective service workers (*t* = −0.64 and *p* = 0.52). Age showed no correlation with psychological impact except for media professionals, where younger workers had higher IES-R scores (Pearson’s *r* = −0.28 and *p* < 0.001). Marital status was not associated with IES-R scores (*p* > 0.05 in all cases). Educational level was inversely related to the IES-R scores in healthcare workers (Spearman’s *ρ* = −0.13 and *p* = 0.03) and grocery workers (Spearman’s *ρ* = −0.23 and *p* = 0.03), while it was unrelated in media professionals and protective service workers. The number of children was directly and weakly correlated with psychological impact only in healthcare workers (Spearman’s *ρ* = 0.153 and *p* = 0.012).

The regions that were affected the most by COVID-19 in Spain during the data collection period were Comunidad de Madrid, Castilla La Mancha, and Cataluña [49]. Therefore, we grouped our sample into two groups: most affected (*n* = 296) and least affected (*n* = 250). We only found differences in the healthcare workers group, where those professionals in the most affected regions showed higher psychological impacts (*M* = 0.38, *SD* = 0.81) than those living in the less affected areas (*M* = 0.14, *SD* = 0.88, *t* = −2.28, and *p* = 0.02). To analyze whether there were any differences in psychological impact, depending on the moment on which the participants completed the questionnaires, we divided the sample into three groups—those who participated during March (*n* = 294), those who participated during April (*n* = 140), and those who participated during May–June (*n* = 112). Differences were then tested with ANOVAs for each group of professionals. While there seemed to be an upward trend as time passed by in healthcare providers, protective service workers, and media professionals, the opposite tendency was found in the case of grocery workers. However, none of these differences were revealed to be significant (*p* > 0.05). It was close to significance in the case of grocery workers (*F* = 3.20 and *p* = 0.05), who showed lower psychological impacts in May–June (*M* = 36.88 and *SD* = 19.60) than in March (*M* = 47.82 and *SD* = 18.77) and April (*M* = 45.72 and SD = 16.15), but the post-hoc Tuckey test revealed no significant differences.

### 3.2. Degree of Concern

The three main concerns for all the professional groups were about the contagion and the uncertainty about the crisis: the possibility of infecting a beloved one with COVID-19, becoming infected by COVID-19, and not knowing when this crisis was going to come to an end (see Table 3). Associations of each concern with the IES-R scores showed that, in general terms, higher levels of concern were associated with higher scores of psychological impact in the four samples. Concretely, the concerns that showed the highest correlations with the IES-R scores were: “Not being emotionally prepared to face my work during the pandemic,” “How much this situation can affect me psychologically,” and “The degree of pressure and stress I am facing at work.” Finally, the ANOVAs revealed that the biggest group differences were found, on the one hand, in “Becoming infected by coronavirus at work,” where grocery workers scored the highest (followed by healthcare workers and protective service workers), while media professionals were much less concerned about it. On the other hand, the concern “Infecting a beloved one by coronavirus” worried grocery workers the most, followed by healthcare workers, protective service workers, and media workers.

### 3.3. Work-Related Variables

The descriptive data of the common work-related variables and their associations with psychological impact can be found in Table 4. Nearly a third of the workers within all professional groups except for grocery workers had experience in working during a similar crisis: healthcare workers at 40.5%, protective service workers at 34.9%, and media professionals at 32.4%. However, both this prior experience and the number of working hours per week were unrelated to the psychological impact they reported. Notably, the protection equipment was reported as insufficient by all the professional groups except by grocery workers and this correlated with significantly higher psychological impact in the case of healthcare workers. Professionals who reported overworking during the crisis (60% of media professionals, 38.3% of healthcare providers, 38.2% of grocery workers, and 21.7% of protective service workers) showed higher psychological impacts, except in the case of protective service workers. The overworking time surpassed 6 h in all professions, with a peak of 11.64 h in the healthcare group. However, the average number of extra hours was not related to the participants’ psychological impact. The years of experience was only significantly and inversely related to the psychological impact in media professionals. The number of days working during the pandemic and the number of days since their last day off were significantly and inversely related to psychological impact. The time spent with other people at work was not related to psychological impact. Finally. the feeling of teamwork was high in all the professional groups (reporting a mean of 2 out of 3), and it was associated with lower psychological impacts only in grocery workers.

The descriptive data of the work-related variables specific to each subsample and their association with psychological impact are included in Table 5. Starting with healthcare workers, the physicians showed lower psychological impacts than the nursing personnel and other healthcare positions. More than a quarter of the healthcare workers (27.1%) had switched from their unit to a COVID-specific one during the pandemics, but this was not associated with their psychological impact. Their health state regarding COVID-19 was also unrelated to psychological impact, although 62.8% ignored whether they were infected because they had never had a COVID-19 test. The patients’ health state was a relevant factor for the healthcare workers’ psychological impact: taking care of dying patients with COVID-19 (43.9%) was strongly associated with higher psychological impact, as was working with patients with COVID-19 (77%). Regarding grocery, media, and protective service workers, none of the specific work-related variables we measured were associated with psychological impact.

### 3.4. Support Received by Their Company/Workplace during the COVID-19 Crisis

Table 6 shows the descriptive data about the support provided by the hiring company in each group of frontline workers and their association with psychological impact. Surprisingly, over half of the healthcare professionals and the protective service workers did not receive any support, while over 60% of media professionals and grocery workers reported one at least type of support. Conciliation was the type of support most reported by the four groups of professionals, ranging from 20% to 40%. Psychological support was mostly received by the healthcare professionals (around a quarter), followed by 11% of the protective service professionals. Finally, the economic support was a frequent reward only in the grocery workers group (around 40%), and it was inversely related to psychological impact in the case of the healthcare professionals.

### 3.5. Perceived Social Recognition and Self-Assessment of the Importance of Their Work during the Pandemic

As Table 7 shows, the participants though that society recognized their work during the pandemic (means ranging between 1.41 and 1.64 within a range of 0–3), without significant differences among professional groups after checking for ANOVA’s post hoc tests. A higher perceived social recognition was related to lower psychological impact in all groups but media professionals. The assessment of the relevance of their own work was high (means ranging between 2.21 and 2.44 within a range of 0–3), with significant differences between healthcare workers and media professionals. The former group considered their work more important than the latter; for healthcare workers, such importance was related to higher psychological impact, though weakly.

### 3.6. Perceived Severity of the Situation

All the groups perceived the crisis as very severe (all scores were over 8.8 out of 10), but there were differences among groups of profession: grocery workers showed a higher perceived severity than media professionals and protective service workers (*p* = 0.03 for both; see Table 7). Importantly, a higher level of perceived severity was associated with more psychological impact in healthcare workers.

## 4. Discussion

To our knowledge, this has been the first study in Spain that evaluated the psychological impact (and the demographic and work-related associated factors) of working during the COVID-19 pandemic in four groups of frontline workers: healthcare, grocery, mass media, and protective professionals. Our main finding is that all four groups evaluated the pandemic situation as severe or very severe, reported feeling sadness, and showed mild to severe psychological impact, with health care and grocery professionals being, by far, the most affected groups and protective professionals being the least negatively affected. The course of this impact, although not significant, seemed to gradually increase as the pandemics progressed, except for the grocery workers, who showed a reverse pattern. It is also worth noting that, despite this general distress, most of the sample reported not having received psychological support.

As a previous study evaluated the psychological impact of the Spanish general population with the same instrument (IES-R) at the beginning of the lockdown [6], we can also report that all frontline workers except the protective service workers showed significantly more psychological impact symptoms than the general population. The protective service professionals’ lower results (that were still high because their scores indicated mild psychological impact) might have been due to the high profile of working-related resilience and hardiness [50,51] or to some assumption of self-imposed stoicism and permanent availability for duty [52]. The worsening labor scenario as time progressed (e.g., more exposure to the virus and accumulative and varying knowledge) might have accounted for the relation between time and higher psychological impact in the professional groups that had to provide individual assistance or handle information, whereas the reverse pattern found in grocery markets might be explained by an initial shock that was replaced by a progressive normalization of their new working functions as restrictions were gradually lifted. In this sense, occupational health during crisis periods would benefit from longitudinal studies with bigger samples to explore the temporal evolution of the workers’ psychological impact.

The demographic and work-related variables that correlated with higher psychological impacts were lower educational level, female gender, considering the crisis as very severe, a lack of sufficient individual protection equipment, being afraid of infecting or becoming infected with COVID-19, uncertainty about the future, and subjective feelings within the work environment such as overworking, a low recognition of the work done, and not being emotionally prepared to cope with such a working scenario. Interestingly, previous working experience in critical situations did not appear to be a protective factor against psychological impact, which contradicts results from previous studies [21,22].

Some of the variables mentioned above may have accounted for the high psychological impact found in healthcare and grocery professionals. First, grocery workers had a lower educational level than the other groups and recognized not having the training needed to face this crisis, which may have increased their uncertainty and distress to face such a new shift [21]. Other COVID-19 studies have also found lower educational levels to correlate with higher psychological symptoms (e.g., [46,53]). Second, both grocery and healthcare workers are highly feminized professions, and evidence has exposed a clear gender gap in the number of contagions (76% women, [54]). Third, being female has repeatedly proved to be a risk factor for developing mental disorders when facing the COVID-19 crisis, both in the general population [6] and in healthcare professionals [24,53]. Lastly, the extremely reduced nurse/patient ratio in Spain [55] would have forced these female professionals to overwork.

Indeed, many professionals considered that they were overworking during the pandemic (more than 38% of healthcare and grocery workers, 60% of media professionals, and 21.7% of protective service workers). Coherently, all the groups reported an average of more than six extra working hours the previous week, with the healthcare group showing the highest mean (*M* = 11.64). The perception of overworking was associated with higher psychological impact in all frontline workers’ mental health, except for protective service workers. However, the number of extra hours did not correlate with higher psychological impacts. This result shows that it is not the objective number of extra hours worked but the subjective feeling of overworking that influences mental health. For example, some media professionals could be used to stay permanently updated, but during the coverage of the pandemic, the flow of information was constant, varying, and with such potential amount of misinformation and fake news that it was considered an infodemic [56]. Additionally, healthcare workers may not only have surpassed their regular work shifts but also have faced an unprecedent sustained overload of higher risk of becoming infected and higher moral injury or distress than usual (e.g., [57,58]). The situation seems to have an increased impact when healthcare workers assist to dying patients. Moreover, those who reported not feeling emotionally prepared to cope with the new working scenario and were concerned about how the situation would affect them psychologically scored higher on the IES-R. In general terms, participants considered that their work was moderately recognized by the society, with no differences between groups. Perceiving higher social recognition was associated with lower psychological impacts, a result also present in other studies (e.g., [59]). Consequently, the symbolic gestures of public recognition towards frontline workers (such as scheduled applauses or dedicated advertisements) could potentially contribute to alleviate psychological distress if they are proven to improve the perception of social recognition of professionals’ work.

The fear of infection is a concern commonly found in the literature about frontline workers during the COVID-19 crisis (e.g., [16]), and this fear was associated with higher psychological impact in our study. The threat of infection is real. For example, around 24% of all the Spanish COVID-19 cases were healthcare workers [54,60], and these professionals showed the highest seroprevalence rate among all occupations in Spain [34]. Not having the adequate protective material was a global problem, and the concern about this defenselessness contributed to higher psychological impacts; therefore, strategies to assure the provisioning of protective material seems a must for frontline workers.

We are aware that the present study was not without limitations. First, the four groups selected do not represent all the possible frontline professionals working during the COVID-19 pandemic. By selecting only healthcare, grocery, mass media, and protective professionals, we missed other relevant frontline workers such as those related to transportation services or finances. Second, the sample sizes were not equal in all the groups. Less than 100 workers participated in the groups of grocery workers and protective service professionals. Finally, the online recruitment may have influenced data collection, since it may have discouraged older workers to participate in the study. Due to the above-mentioned reasons, the findings of this study should only be generalized with caution.

The results of this study can help make recommendations aimed at protecting mental health in frontline workers. First, special attention should be paid to healthcare workers, grocery workers, and media professionals, regardless of their previous experience in similar crises. Specifically, dedicated attention should be paid to females working in the healthcare and grocery sectors and to young media and protective service workers. Within healthcare workers, the most vulnerable professionals are those working in the most affected regions, tending to COVID-19 patients, and/or taking care of dying patients with COVID-19, so more intensive monitoring and measures might be applied to them. Second, acknowledging the potential differences between professional groups regarding how the progression of the crisis impacts their working conditions might also help to provide timely support. In line with other authors [61,62] and in the light of our results, a relevant message for human resource departments is the necessity of periodically monitoring the psychological health of frontline workers and acting accordingly.

Third, human resource departments should also schedule frequent breaks for their employees and check and improve their workflows, as well as make sure that their employees regularly enjoy days off in order to reduce objective and subjective overload. Additionally, personal protection equipment should be provided to all frontline workers, as previously stated, to reduce their concerns about becoming infected or infecting others and thus lowering the psychological impact of the crisis. Fourth, frontline workers should also be provided with emotional training on emergency coping techniques by their employers, and easy access to psychological care should be provided to them or at least facilitated [61]. In this sense, educating healthcare professionals on palliative care and grief management would help to alleviate the psychological impact of working with dying patients in such a difficult situation. Finally, measures should be taken whenever possible to increase the perceived social recognition of their work at both the formal governmental level and the informal level.

Strategically, a national network of psychological support with crisis intervention expertise should be integrated in the human resource departments of frontline workers’ institutions or, more generally, in the National Health Provision—at least during crisis periods—to provide specialized support and prevent the appearance of mental health issues. Importantly, psychologists providing crisis intervention should be proactive instead of waiting to be asked for help. This is essential to reach professionals that may hide their emotional state (such as protective workers in our sample) or those who traditionally delay their request for assistance (such as health workers) [61].

## 5. Conclusions

In conclusion, our results show that the proportion of frontline workers showing high psychological impacts derived from working during the COVID-19 sanitary crisis is alarmingly high. This can be explained by the severity of the crisis but also by the failure of the hiring companies and the national health strategy to provide workers the necessary support. Indeed, despite the severity of the psychological impacts and the indicators of sadness in all frontline workers, psychological assistance was surprisingly absent for 90% of the sample. Therefore, letting psychologists who are experts on crisis management help heads of sections customize labor measures (as was done in China; see [9]) by selecting the concerns that correlate with high psychological distress—and doing so considering the course of psychological impact—would allow for hiring companies to screen their workers’ psychological health and quickly figure out which actions to take to prevent the development of negative symptoms.

## Figures and Tables

**Table 1 ijerph-17-08149-t001:** Sociodemographic characteristics of the four subsamples.

	Healthcare Workers	Media Professionals	Grocery Workers	Protective Service Workers
	*N* (%)	*N* (%)	*N* (%)	*N* (%)
**Gender**				
Male	55 (20.4)	50 (47.6)	17 (19.1)	71 (85.5)
Female	212 (78.8)	55 (52.4)	72 (80.9)	12 (14.5)
Rather not to say	2 (0.7)			
**Marital Status**				
Married/cohabiting	162 (60.2)	54 (51.4)	58 (65.2)	61 (73.5)
Single	68 (25.3)	46 (43.8)	25 (28.1)	17 (20.5)
Separated/divorced	38 (14.1)	4 (3.8)	6 (6.7)	5 (6)
Widow(er)	1 (0.4)	1 (1)	0	0
**Nº of Children**				
No children	173 (64.3)	74 (70.5)	42 (47.2)	39 (47)
One	41 (15.2)	14 (13.3)	21 (23.6)	17 (20.5)
Two	42 (15.6)	15 (14.3)	22 (24.7)	22 (26.5)
Three or more	13 (4.8)	2 (1.9)	4 (4.5)	5 (6)
**Education Level**				
Primary education	1 (0.4)	0	2 (2.2)	0
Secondary compulsory education	1 (0.4)	0	21 (23.6)	7 (8.4)
Secondary post-compulsory education	5 (0.19)	0	14 (15.7)	21 (25.3)
Professional training	37 (13.8)	9 (8.6)	28 (31.5)	12 (14.5)
University degree	173 (64.3)	45 (42.6)	18 (20.2)	27 (32.5)
Master’s degree	36 (13.4)	44 (41.9)	6 (6.7)	13 (15.7)
Ph.D.	0	7 (6.7)	0	3 (3.6)
**Age**				
	*M* (*SD*)	*M* (*SD*)	*M* (*SD*)	*M* (*SD*)
	39.77 (11.34)	39.22 (10.98)	38.04 (8.66)	40.17 (8.62)

Note. *M* = Mean.

**Table 2 ijerph-17-08149-t002:** Psychological impact (Impact of Event Scale-Revised (IES-R) standardized factorial score) of working during the COVID-19 pandemic, indicators of sadness, and ANOVAs results testing for differences between groups of profession. PI: psychological impact.

	Healthcare Workers	Media Professionals	Grocery Workers	Protective Service Workers	ANOVAs		
**Psychological Impact of the COVID-19 Pandemic (IES-R)**							
	*n* (%)	*n* (%)	*n* (%)	*n* (%)			
Minimal PI	24 (8.9)	21 (20)	17 (19.1)	45 (54.2)			
Mild PI	35 (13)	27 (25.7)	10 (11.2)	11 (13.3)			
Moderate PI	12 (4.5)	6 (5.7)	4 (4.5)	5 (6)			
Severe PI	198 (73.6)	51 (48.6)	58 (65.2)	22 (26.5)			
	*M* (*SD*)	*M* (*SD*)	*M* (*SD*)	*M* (*SD*)	*F*	*p* *	η^2^
IES-R standardized factorial score	0.28 (0.84) ^a^	−0.22 (0.87) ^b^	0.15 (1.07) ^a^	−0.80 (0.84) ^c^	34.47	<0.001	0.16
**Feeling Depressed (Are You Depressed?)**							
	*n* (%)	*n* (%)	*n* (%)	*n* (%)			
No	135 (50.2)	66 (62.9)	39 (43.8)	68 (81.9)			
Maybe	96 (35.7)	31 (29.5)	37 (41.6)	13 (15.7)			
Yes	38 (14.1)	8 (7.6)	13 (14.6)	2 (2.4)			
**Level of Sadness**							
	*M* (*SD*)	*M* (*SD*)	*M* (*SD*)	*M* (*SD*)	*F*	*p* *	η^2^
Sadness (0–10)	6.34 (2.45) ^a^	5.35 (2.52) ^b,c^	5.87 (2.67) ^a,b^	4.55 (2.61) ^c^	11.96	<0.001	0.06

Note. In order to calculate the number and percentage of participants on each range of the IES-R, the IES-R direct score and cited cutoff criteria were used. For the rest of analyses, the IES-R standardized factorial scores were used. * Frontline professionals with a different superscript letter in the same row show a significant difference between them in the variable of that row. The effect size was assessed via η^2^ (interpretation: negligible < 0.01 < small < 0.06 < medium < 0.14 < large).

**Table 3 ijerph-17-08149-t003:** Average level of concern in all the groups, the Spearman’s correlation (ρ) with psychological impact (IES-R standardized factorial score) and ANOVA results testing for differences between groups of profession.

Concerns (Likert 0 to 3)	Healthcare Workers		Media Professionals		Grocery Workers		Protective Service Workers		ANOVAs		
**Common Concerns for All Professions**	*M* (*SD*)	*ρ*	*M* (*SD*)	*ρ*	*M* (*SD*)	*ρ*	*M* (*SD*)	*ρ*	*F*	*p*	η^2^
Not being emotionally prepared to face my work during the pandemic	1.58 (0.86) ^a^	0.42 ***	1.12 (0.86) ^b,c^	0.49 ***	1.39 (0.91) ^a,b^	0.65 ***	0.89 (0.86) ^c^	0.48 ***	16.25	<0.001	0.08
Not having enough training to face my current work during the pandemic	1.50 (0.84) ^a,c^	0.074	0.88 (0.76) ^b^	0.19 *	1.11 (0.91) ^b,d^	0.34 ***	1.37 (1) ^c,d^	0.14	14.98	<0.001	0.08
Not being up to the events in my work in the current situation	1.51 (0.90) ^a^	0.23 ***	1.09 (0.89) ^b^	0.35 ***	1.04 (0.89) ^b^	0.38 ***	0.93 (0.87) ^b^	0.22 *	14.12	<0.001	0.07
Becoming infected by coronavirus at work	2.02 (0.88) ^a^	0.40 ***	1.19 (1.26) ^1 b^	0.21 * ^1^	2.36 (0.80) ^c^	0.43 ***	2.05 (0.75) ^a^	0.41 ***	28.97	<0.001	0.14
The situation of collective nervousness at work	1.94 (0.80) ^a^	0.35 ***	1.40 (1) ^2 b^	0.28 ** ^2^	1.74 (0.89) ^a,b^	0.43 ***	1.69 (0.73) ^b^	0.39 ***	9.62	<0.001	0.05
Infecting a beloved one by coronavirus	2.72 (0.53) ^a^	0.23 ***	2.13 (0.96) ^b^	−0.03	2.69 (0.63) ^a^	0.22 *	2.76 (0.51) ^a^	0.27 *	23.54	<0.001	0.12
Not knowing when this crisis is going to come to an end	2.37 (0.70) ^a^	0.36 ***	2.17 (0.77) ^a,b^	0.45 ***	2.43 (0.77) ^a^	0.55 ***	2.02 (0.70) ^b^	0.12	6.75	<0.001	0.04
How much this situation can affect me psychologically	1.97 (0.79) ^a^	0.54 ***	1.71 (0.89) ^a^	0.73 ***	1.93 (1.02) ^a^	0.70 ***	1.25 (0.81) ^b^	0.55 ***	15.90	<0.001	0.08
The degree of pressure and stress I am facing at work	2.00 (0.80) ^a^	0.52 ***	1.76 (0.84) ^a^	0.56 ***	2.04 (0.90) ^a^	0.61 ***	1.27 (0.86) ^b^	0.52 ***	18.33	<0.001	0.09
My family is concerned about me because I am working during the pandemic	2.27 (0.71) ^a^	0.40 ***	1.49 (0.98) ^b^	0.37 ***	2.24 (0.85) ^a^	0.45 ***	2.17 (0.64) ^a^	0.12	26.73	<0.001	0.13
**Specific Concerns for Healthcare workers**	*M* (*SD*)	*r*									
Not being able to dedicate enough time or to attend every patient due to excessive burden or equipment shortage	2.26 (0.80)	0.27 ***									
Not being able to provide the necessary emotional support to patients and families	2.28 (0.74)	0.23 ***									
Lack of clinical information about the virus	2.10 (0.84)	0.37 ***									
**Specific Concerns for Media Professionals**			*M* (*SD*)	*r*							
Communicating fake information or making mistakes			2.01 (0.81)	0.17							
The psychological impact that the information I generate or contribute to spread may cause in the population			2.03 (0.83)	0.11							
Not being able to keep myself constantly updated			1.77 (0.91)	0.28 **							
Not to handle the information properly, because I am receiving it at a higher rate than usual			1.94 (0.86) ^3^	0.33 ***							
**Specific Concerns for Protective Service Workers**							*M* (*SD*)	*r*			
Not being able to make people law-abiding							1.87 (0.81)	−0.07			

Note. Frontline professionals with a different superscript letter in the row show a significant difference between them in the variable of that row. The effect size was assessed via η^2^ (interpretation: negligible < 0.01 < small < 0.06 < medium < 0.14 < large). ^1^
*N* = 17 participants were excluded from this analysis because they indicated “It does not apply to me, because I telework.” ^2^
*N* = 13 participants were excluded from this analysis because they indicated “It does not apply to me, because I telework.” ^3^
*N* = 6 participants were excluded from this analysis because they indicated “It does not apply to me, because I play a technical role.” * *p* < 0.05. ** *p* < 0.01. *** *p* < 0.001.

**Table 4 ijerph-17-08149-t004:** Descriptive data of common work-related variables and association with IES-R standardized factorial score.

	Healthcare Workers					Media Professionals					Grocery Workers					Protective Service Workers				
	*N* (%)	*M* (*SD*)	*t*/*F* *	*p*	*g*/η^2^	*N* (%)	*M* (*SD*)	*t*/*F* *	*p*	*g*/η^2^	*N* (%)	*M* (*SD*)	*t*/*F* *	*p*	*g*/η^2^	*N* (%)	*M* (*SD*)	*t*/*F* *	*p*	*g*/η^2^
**Experience in Previous Crises**																				
No	160 (59.5)	0.27 (0.89)	−0.18	0.86	0.02	71 (67.6)	−0.23 (0.90)	−0.17	0.87	0.03	88 (98.9)	0.17 (1.07)	^1^			54 (65.1)	−0.78 (0.77)	0.30	0.77	0.07
Yes	109 (40.5)	0.29 (0.79)				34 (32.4)	−0.20 (0.81)				1 (1.1)	−1.34				29 (34.9)	−0.84 (0.96)			
**Working Hours Per Week**																				
Less than 10 h	12 (4.5)	0.08 (1.09)	1.55	0.19	0.02	8 (7.6)	0.24 (0.76)	3.43	0.01 ^2^	0.12	8 (9.0)	−0.25 (1.17)	1.19	0.32	0.05	4 (4.8)	−1.33 (0.51)	1.34	0.26	0.06
10–20 h	8 (3.0)	−0.19 (0.55)				8 (7.6)	0.16 (0.80)				7 (7.9)	0.63 (1.22)				2 (2.4)	−0.69 (1.06)			
20–30 h	26 (9.7)	0.53 (0.81)				8 (7.6)	−0.82 (0.54)				24 (27.0)	0.42 (0.97)				7 (8.4)	−0.86 (0.81)			
30–40 h	108 (40.2)	0.23 (0.82)				37 (35.2)	−0.48 (0.81)				32 (36.0)	0.01 (1.10)				55 (66.3)	−0.86 (0.85)			
Over 40 h	115 (42.8)	0.33 (0.85)				44 (41.9)	−0.05 (0.90)				18 (20.2)	0.04 (1.05)				15 (18.1)	−0.41 (0.80)			
**Availability of Personal Protection Equipment**																				
No	7 (2.6)	0.69 (0.59) ^a^	4.38	0.01	0.03	17 (16.2)	−0.04 (0.93)	1.04	0.38	0.03	3 (3.4)	.14 (0.89)				16 (19.3)	−0.64 (0.95)	2.77	0.07	0.07
Yes, but not enough	184 (68.4)	0.33 (0.79) ^a,b^				17 (16.2)	−1.4 (0.89)				27 (30.3)	0.46 (1.16)	1.81	0.07 ^4^	0.42	55 (66.3)	−0.73 (0.82)			
Yes, enough	78 (29)	0.11 (0.95) ^b^				16 (15.2)	−0.54 (0.75)				59 (66.3)	0.01 (1.02)				12 (14.5)	−1.30 (0.55)			
I telework full-time						55 (52.4)	−0.21 (0.87)													
**Overworking during COVID-19 Crisis**																				
No	166 (61.7)	0.20 (0.83)	−2.06	0.04	0.26	42 (40.0)	−0.46 (0.86)	−2.38	0.02	0.47	55 (61.8)	−0.05 (1.03)	−2.32	0.02	0.51	65 (78.3)	−0.84 (0.85)	−0.96	0.34	0.25
Yes	103 (38.3)	0.42 (0.86)				63 (60.0)	−0.06 (0.84)				34 (38.2)	0.48 (1.07)				18 (21.7)	−0.63 (0.80)			
		*M* (SD)	*r*/*ρ* **	*p*			*M* (SD)	*r*/*ρ* **	*p*			*M* (SD)	*r*/*ρ* **	*p*			*M* (SD)	*r*/*ρ* **	*p*	
**Extra Hours Last Week**		11.64 (9.34)	0.11	0.28			6.10 (4.28)	0.25	0.06			7.80 (7.24)	−0.11	0.56			8.83 (7.09)	0.17	0.50	
**Years of Experience**		14.87 (10.85)	0.01	0.81			15.88 (10.97)	−0.27	<0.01			13.44 (8.24)	−0.10	0.34			16.29 (11.39)	0.01	0.96	
**Days Working during the Pandemic**		20.13 (20.73)	0.04	0.55			13.18 (12.26)	0.17	0.08			31.22 (20.53)	−0.23	0.03			12.35 (10.13)	0.10	0.35	
**Days since Last Day Off**		2.42 (3.60)	0.05	0.42			3.81 (4.06)	0.14	0.16			2.66 (2.94)	−0.33	< 0.01			1.84 (2.45)	−0.08	0.47	
**Working as a Team**		2.34 (0.77)	−0.02	0.80			2.03 (0.96)	−0.06	0.55			2.07 (0.88)	−0.24	0.02			2.21 (0.80)	−0.20	0.08	
**Time in Contact with People ^3^**							2.11 (1.68)	0.00	0.97			3.48 (1.19)	0.03	0.83			3.47 (0.89)	0.14	0.22	

* Differences in mean level between categories of dichotomous variables were assessed via *t*-tests, and Hedges’ *g* effect size statistic was obtained (interpretation: negligible < 0.20 < small < 0.50 < medium < 0.80 < large). For multiple-category variables, one-way ANOVAs were used, and categories with a different superscript letter show a significant difference between them in the psychological impact variable’s mean. In these cases, the effect size was assessed via η^2^ (interpretation: negligible < 0.01 < small < 0.06 < medium < 0.14 < large). ** Correlations with ordinal variables (e.g., working as a team and time in contact with people) were computed via Spearman’s correlation (*ρ*). Correlations with continuous variables were computed via Pearson’s correlation (*r*). The obtained statistics are themselves measures of effect size (interpretation: negligible < 0.10 < small < 0.30 < medium < 0.50 < large). ^1^ Analysis not performed because one group had only one case. ^2^ Tukey post-hoc analyses revealed no differences. ^3^ In healthcare providers, this question was not included because it was assumed that they are necessarily in constant touch with others. ^4^ As the first group had only 3 participants, it was excluded from the analysis, and a *t*-test between the two remaining groups was carried out.

**Table 5 ijerph-17-08149-t005:** Descriptive data of specific work-related variables and their association with psychological impact (IES-R standardized factorial score).

Healthcare Workers						Media Professionals					
	*N* (%)	*M* (*SD*)	*t*/*F* *	*p*	*g*/η^2^ *		*N* (%)	*M* (*SD*)	*t*/*F* *	*p*	*g*/η^2^ *
**Profession**						**Professional Role**					
Physician	87 (32.39)	0.01 (0.79) ^a^	7.39	0.001	0.05	Press conferences					
Nurse/nursing assistant	131(48.70)	0.43 (0.77) ^b^				Yes	20 (19)	−0.37 (1)	0.87	0.39	0.21
Other	51 (18.96)	0.36 (1.00) ^b^				No	85 (81)	−0.19 (0.83)			
**Current Situation**						Live communications					
Working	250 (92.9)		^1^			Yes	20 (19)	−0.18 (0.86)	−0.24	0.82	0.06
Quarantined (COVID)	14 (5.2)					No	85 (81)	−0.23 (0.87)			
On leave (other reason)	5 (1.9)					Interviews					
**Change of Unit during COVID-19**						Yes	38 (36.2)	−0.18 (0.79)	−0.35	0.73	0.07
Yes	73 (27.1)	0.18 (0.80)	1.22	0.23	0.17	No	67 (63.8)	−0.24 (0.91)			
No	196 (72.9)	0.32 (0.86)				Covering breaking news					
**COVID-19 Test**						Yes	56 (53.3)	−0.16 (0.87)	−0.81	0.42	0.16
Yes, positive result	26 (9.7)	0.39 (0.93)	0.27	0.77	0.00	No	49 (46.7)	−0.30 (0.87)			
Yes, negative result	69 (25.7)	0.25 (0.90)				Writing reports					
No	169 (62.8)	0.28 (0.85)				Yes	57 (54.3)	−0.12 (0.83)	0.67	0.17	0.27
I would rather not to say	5 ^2^ (1.9)	0.11 (0.51)				No	48 (45.7)	−0.35 (0.90)			
**Attending Patients with COVID-19**						Social media					
Yes	208 (77.3)	0.36 (0.82)	−2.66	0.01	0.40	Yes	31 (29.5)	−0.23 (0.86)	−0.21	0.83	0.05
No	61 (22.7)	0.03 (0.88)				No	74 (70.5)	−0.19 (0.89)			
**Attending Severe Patients with COVID-19**						Technical assistance					
Yes	140 (52)	0.36 (0.82)	−1.50	0.14	0.19	Yes	14 (13.3)	−0.29 (0.93)	0.32	0.75	0.09
No	129 (48)	0.20 (0.86)				No	91 (86.7)	−0.21 (0.86)			
**Attending Dying Patients with COVID-19**						Teleprinter					
Yes	118 (43.9)	0.49 (0.83)	−3.62	<0.001	0.45	Yes	12 (11.4)	−0.27 (1)	0.19	0.85	0.07
No	151 (56.1)	0.12 (0.82)				No	93 (88.6)	−0.21 (0.85)			
**Grocery Workers**						Technical assistance on the street					
**Working Role**						Yes	10 (9.5)	0.11 (0.89)	−1.28	0.21	0.43
Cashiers/stock replenisher	49 (55.5)	0.38 (1.1)	^3^			No	95 (90.5)	−0.26 (0.86)			
Transportation/delivery	5 (5.6)	−0.08 (1)				**Work modality**					
Stalls	4 (4.5)	1.13 (0.56)				Teleworking	73 (69.5)	−0.24 (0.89)	0.31	0.76	0.07
Area manager	6 (6.7)	−0.78 (0.88)				On-site working	32 (30.5)	−0.18 (0.82)			
Human resources	7 (7.9)	0.01 (1.03)				**Media**					
Own store	3 (3.4)	0.41 (1.19)				Written press	8 (7.6)	−0.40 (0.95)	0.25	0.94	0.013
Various roles	15 (16.9)	−0.39 (0.96)				Digital press	32 (30.5)	−0.25 (0.82)			
**Size of the Shop**						Television	18 (17.1)	−0.19 (0.99)			
Corner shop	8 (9)	0.22 (1.11)	2.02	0.15	0.05	Radio	12 (11.4)	−0.37 (0.59)			
Supermarket	55 (61.8)	−0.01 (1.09)				Other	19 (18.1)	−0.08 (1)			
Hypermarket	26 (29.2)	0.49 (0.98)				Various	16 (15.2)	−0.18 (0.83)			
**Protective Service Workers**						**Audience Impact**					
**Unit**						Local	12 (11.4)	−0.25 (0.85)	0.22	0.81	0.004
Civil guard	47 (56.6)	−0.78 (0.91)	0.47	0.71	0.02	Regional	30 (28.6)	−0.30 (1)			
Local police	11 (13.3)	−0.83 (0.51)				National	63 (60)	−0.18 (0.80)			
National police	17 (20.5)	−0.68 (0.77)									
Army	8 (9.6)	-1.10 (9.1)									
**Working Role**											
Patrolling											
Yes	56 (67.5)	−0.76 (0.84)	−0.61	0.54	0.13						
No	27 (32.5)	−0.87 (0.83)									
Providing information											
Yes	17 (20.5)	−0.82 (0.99)	0.14	0.89	0.04						
No	66 (79.5)	−0.79 (0.80)									
Disinfection											
Yes	4 (4.8)	−0.60 (1.30)	^4^								
No	79 (95.2)	−0.81 (0.82)									
Guard critical structures											
Yes	16 (19.3)	−0.64 (1.08)	−0.82	0.42	0.23						
No	67 (80.7)	−0.83 (0.77)									
Border or road control											
Yes	20 (24.1)	−0.88 (0.99)	0.50	0.62	0.13						
No	63 (75.9)	−0.77 (0.79)									
Command and coordination											
Yes	19 (22.9)	−0.97 (0.79)	1.01	0.32	0.27						
No	64 (77.1)	−0.74 (0.85)									

* Differences in mean level between categories of dichotomous variables were assessed via *t*-tests, and Hedges’ *g* effect size statistic was obtained (interpretation: negligible < 0.20 < small < 0.50 < medium < 0.80 < large). For multiple-category variables, one-way ANOVAs were used, and categories with different superscript letters show a significant difference between them in the psychological impact variable’s mean. In these cases, the effect size was assessed via η^2^ (interpretation: negligible < 0.01 < small < 0.06 < medium < 0.14 < large). ^1^ This ANOVA was not conducted given that only one group had more than 15 participants. ^2^ This group was not included in the ANOVA (only 5 participants) ^3^ This ANOVA was not conducted given that only two out of seven groups had more than 7 participants. ^4^ This *t*-test was not conducted given that one group only had 4 participants.

**Table 6 ijerph-17-08149-t006:** Support received by the hiring company during the COVID-19 crisis and association with psychological impact (IES-R standardized score).

	Healthcare Workers					Media Professionals					Grocery Workers					Protective Service Workers				
	*N* (%)	*M* (SD)	*t*	*p*	*g*	*N* (%)	*M* (SD)	*t*	*p*	*g*	*N* (%)	*M* (SD)	*t*	*p*	*g*	*N* (%)	*M* (SD)	*t*	*p*	*g*
**Any support**																				
No	135 (50.2)	0.32 (0.82)	0.81	0.42	0.09	34 (32.4)	−0.13 (1.03)	0.72	0.47	0.16	35 (39.3)	0.25 (1.01)	0.68	0.50	0.15	44 (53.0)	−0.79 (0.83)	0.09	0.93	0.02
Yes	134 (49.8)	0.24 (0.87)				71 (67.6)	−0.27 (0.78)				54 (60.7)	0.09 (1.12)				39 (47.0)	−0.81 (0.85)			
**Psychological**																				
No	194 (72.1)	0.29 (0.85)	0.23	0.82	0.04	102 (97.1)	−0.21 (0.86)	0.66	0.51	0.39	88 (98.9)	0.16 (1.07)	^1^			74 (89.2)	−0.81 (0.81)	−0.28	0.78	0.11
Yes	75 (27.9)	0.26 (0.84)				3 (2.9)	−0.55 (1.32)				1 (1.1)	−0.03 (-)				9 (10.8)	−0.72 (1.08)			
**Conciliation**																				
No	215 (79.9)	0.32 (0.82)	1.51	0.13	0.23	83 (79.0)	−0.21 (0.90)	0.28	0.78	0.07	63 (70.8)	0.09 (1.07)	0.90	0.37	0.21	50 (60.2)	−0.73 (0.85)	−0.93	0.35	0.20
Yes	54 (20.1)	0.13 (0.91)				22 (21.0)	−0.27 (0.73)				26 (29.2)	0.32 (1.09)				33 (39.8)	−0.90 (0.81)			
**Economical**																				
No	261 (97.0)	0.30 (0.84)	2.33	0.02	0.83	104 (99.0)	−0.23 (0.87)	^1^			54 (60.7)	0.27 (1.00)	1.32	0.19	0.28	83 (100)	−0.80 (0.83)	^1^		
Yes	8 (3.0)	−0.40 (0.86)				1 (1.0)	0.17 (-)				35 (39.3)	−0.03 (1.16)				0	-			

Note. Differences in mean levels between categories of dichotomous variables were assessed via *t*-tests, and Hedges’ *g* effect size statistic was obtained (interpretation: negligible < 0.20 < small < 0.50 < medium < 0.80 < large). ^1^ Analysis not performed because one group had only one case or had no cases.

**Table 7 ijerph-17-08149-t007:** Average levels of perceived social recognition, perceived importance of the own work during the pandemic, perceived severity of the COVID-19 crisis, and their associations with psychological impact (IES standardized factorial score), and differences between groups.

	Healthcare Workers		Media Professionals		Grocery Workers		Protective Service Workers		ANOVAs		
	*M* (*SD*)	*ρ*/*r*	*M* (*SD*)	*ρ*/*r*	*M* (*SD*)	*ρ*/*r*	*M* (*SD*)	*ρ*/*r*	*F*	*p*	η^2^
Perceived social recognition of their work ^1^	1.63 (0.88)	−0.18 **	1.41 (0.74)	−0.08	1.42 (0.81)	−0.29 **	1.64 (0.82)	−0.25 *	2.84	0.04 ^3^	0.02
Perceived importance of their work ^1^	2.44 (0.62) ^a^	0.14 *	2.21 (0.72) ^b^	0.03	2.36 (0.57) ^a,b^	−0.06	2.37 (0.56) ^a,b^	0.11	3.51	0.02	0.02
Perceived severity of the crisis ^2^	9.21 (1.15) ^a,b^	0.18 **	8.90 (1.04) ^b^	−0.05	9.35 (0.96) ^a^	−0.05	8.87 (1.31) ^b^	0.19	4.56	<0.01	0.03

Note. Frontline professionals with different superscript letters in the same row show a significant difference between them in the variable of that row. The effect size was assessed via η^2^ (interpretation: negligible < 0.01 < small < 0.06 < medium < 0.14 < large). ^1^ Spearman’s correlation was computed. ^2^ Pearson’s correlation was computed. ^3^ Tukey post-hoc analyses revealed no differences. * *p* < 0.05; ** *p* < 0.01.

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
