# Peer review of "Working in the Times of COVID-19. Psychological Impact of the Pandemic in Frontline Workers in Spain"

_ijerph, 2020, doi:10.3390/ijerph17218149_

Round 1

Reviewer 1 Report

This paper addresses the important issue of the psychological impact of the COVID-19 pandemic on workers in direct contact with infected or potentially infected people.
The issue raised in the paper is often overlooked in public debate, where the main focus is on people with covid-19.

Comments:
1. Method of selecting the sample. The authors used snowball sampling, which is a nonprobability sampling technique. Consequently, although the authors used appropriate statistical tests, the conclusions of the study cannot be generalized to the entire population under study.
The authors are aware of this limitation, but believe that the conclusions are so important that they justify the use of such a simplified approach to sampling.
2. Time of data collection. The data for the study was collected from March 21 to June 4, 2020. Two and a half months is a fairly short time for data collection, and usually the time effect can be neglected. However, in this particular case, the changes in the intensity of the epidemic were very dynamic. Also, the amount of coverage of covid-19 reported by the media was increasing and more dramatic. Therefore, authors should at least check if there are any differences depending on the period of data collection.

Spelling mistake? Rows 100-102:
"Concretely, among police workers, patrolling is the main source of anxiety [32], but punctual severe crises may cause a myriad of psychological disorders."
punctual severe crises?

Author Response

REVIEWER 1

  1. Method of selecting the sample. The authors used snowball sampling, which is a nonprobability sampling technique. Consequently, although the authors used appropriate statistical tests, the conclusions of the study cannot be generalized to the entire population under study. The authors are aware of this limitation, but believe that the conclusions are so important that they justify the use of such a simplified approach to sampling.

We thank the reviewer for this comment, as it has allowed us to provide more details of the recruitment strategy. It certainly consisted of non-probability sampling techniques, since using a probability sampling technique such as a randomized multi-stage cluster sampling would have required a significantly longer time before being able to begin the data collection, and this would have meant effectively not collecting data of the early moments of the confinement (e.g., March, April), which we believe is actually one of the strengths of our study.

Concerning the non-probability sampling strategies used, there were two, including snowball sampling, but this one was an ancillary technique to the main one. As it is now described more in depth in the procedures section (see lines 230-251) we contacted multiple centers, professional associations, key individuals, etc. regarding each professional group so as to have a wider access to those populations. We hope that the sampling techniques used are clearer now. However, there are of course caveats concerning the generalization of the results, which we have addressed in the discussion section (see lines 520-528).

  1. Time of data collection. The data for the study was collected from March 21 to June 4, 2020. Two and a half months is a fairly short time for data collection, and usually the time effect can be neglected. However, in this particular case, the changes in the intensity of the epidemic were very dynamic. Also, the amount of coverage of covid-19 reported by the media was increasing and more dramatic. Therefore, authors should at least check if there are any differences depending on the period of data collection.

We honestly thank the reviewer for this suggestion. In order to explore the time effect, we divided the sample into three groups: participants who completed the questionnaires in March, participants who completed the questionnaires in April, and those who participated during May-June, and then tested for the differences in the IES-R scores by conducting an ANOVA in each group of professionals (this is described in the results section, lines 300-310). The results showed some patterns, but no significance was reached to assert actual differences. These results and associated implications have been discussed later (see lines 439-441 and 462-469).

3. Spelling mistake? Rows 100-102: "Concretely, among police workers, patrolling is the main source of anxiety [32], but punctual severe crises may cause a myriad of psychological disorders."
punctual severe crises?

Thanks for noticing the mistake. We were referring to one-time, or exceptional crises, that are not frequent stressors, but that given their intensity can cause psychological disorders. Please note that in the text we replace the word punctual with one-time (line 112).

Reviewer 2 Report

The research is related to one of the most urgent and actual issues of current health researches, and I recommend it for publication after minor revision.

The strong point of the research is its actuality, especially considering the surveyed groups - frontline employees in Spain which is one of the World biggest centre of pandemic spreading. Authors developed the methodology of assessment of psychological impact of some extreme situations on workers and apply it to pandemic situation which is quite appropriate for the aim of the research.

However, I have to mention some comments to improve the quality of the research.

(1) The title contains list of groups of "frontline workers". I reckon, these details can be omitted in the title - see notes in the text.

(2) The final statement in the Abstract does not reflect any scientific information and has no link with general text (Given that the proportion of frontline workers showing high PI is alarmingly high, urgent measures are needed). It is obvious even without special scientific research. This statement should be removed. Instead, authors should stress on the aim of the research brief description of the methodology.

(3) Part 2 "Materials and Methods" should be improved in some issues. Authors should describe their logic of samples forming and channels to get the target contacts (see notes in text).

(4) The basic point of authors' research is application of existing methodology for their purposes. In general, it done successfully, however, authors should provide the full list of items of IES-R or define the pages of these two references ([42, 43]) clearly in citations. These sources are unavailable for free reading, so readers cannot "guess" 22 items included in questionnaire. In the given form it is standard references to both chapters with full text, not references to certain pages with IES-R indicators.

(5) One of the drawbacks of the research is uncertainty of the authors' position on scientific novelty and practical application of the results. Thus, if they stress on new valuable approach to measure the psychological impact of pandemic circumstances on workers, they should make appropriate conclusions about scientific soundness of their method and possibilities to use their approach for periodical monitoring, its acceptance in HRM practices and/or on the national level to support such workers which can be developed in related researches. Conclusions done in line 475 - 477 are unconvincing and they narrow the value of the research.

More comments see in the text of the article.

Author Response

Reviewer 2: 

The research is related to one of the most urgent and actual issues of current health researches, and I recommend it for publication after minor revision. The strong point of the research is its actuality, especially considering the surveyed groups - frontline employees in Spain which is one of the World biggest centre of pandemic spreading. Authors developed the methodology of assessment of psychological impact of some extreme situations on workers and apply it to pandemic situation which is quite appropriate for the aim of the research. However, I have to mention some comments to improve the quality of the research.

(1) The title contains list of groups of "frontline workers". I reckon, these details can be omitted in the title - see notes in the text.

We agree with the Reviewer 2 that the previous title was too long. As suggested, now the title is “Working in the times of COVID-19. Psychological impact of the pandemic in frontline workers in Spain”.

(2) The final statement in the Abstract does not reflect any scientific information and has no link with general text (Given that the proportion of frontline workers showing high PI is alarmingly high, urgent measures are needed). It is obvious even without special scientific research. This statement should be removed. Instead, authors should stress on the aim of the research brief description of the methodology.

This general statement has been removed from the abstract and replaced by a more specific conclusion. Also, we have strengthened the description of the methodology by including the variables measured. Please note that after doing these changes (which we think that significantly improves the abstract), we have surpassed the 200 abstract words limit. In the case that this is a problem, we believe that removing the last sentence is the best possible option.

(3) Part 2 "Materials and Methods" should be improved in some issues. Authors should describe their logic of samples forming and channels to get the target contacts (see notes in text).

The Reviewer note in the text reads: The approaches to form the samples should be described more detailed. Why did you choose these 4 groups? Which professional boards were involved to get the contacts?

Thanks for your suggestions. We have described more in depth the procedures section to address the reviewer’s comments (See lines 230-251).

(4) The basic point of authors' research is application of existing methodology for their purposes. In general, it done successfully, however, authors should provide the full list of items of IES-R or define the pages of these two references ([42, 43]) clearly in citations. These sources are unavailable for free reading, so readers cannot "guess" 22 items included in questionnaire. In the given form it is standard references to both chapters with full text, not references to certain pages with IES-R indicators.

We thank the reviewer for this comment. We agree with the need for transparency and free access in science and we indeed strive for it.

The references that the reviewer mentions, #42 and #43, are references that give credit to the original work in English. Please note that since our study was carried out in Spain, we used the Spanish scale by Baguena et al. (2001) (reference #44), which is freely available at https://dialnet.unirioja.es/servlet/articulo?codigo=7061433. We have indicated so in the appropriate reference, adding the URL at which the article is available and the date on which we first accessed it (see lines 707-708).

If the reviewer wishes to access the IES-R scale in English language, the 22 items have been included in the second page of this recent and freely available article: https://hign.org/sites/default/files/2020-06/Try_This_General_Assessment_19.pdf (please note that it only reproduces the scale to facilitate professional use, it is not the original publication).

(5) One of the drawbacks of the research is uncertainty of the authors' position on scientific novelty and practical application of the results. Thus, if they stress on new valuable approach to measure the psychological impact of pandemic circumstances on workers, they should make appropriate conclusions about scientific soundness of their method and possibilities to use their approach for periodical monitoring, its acceptance in HRM practices and/or on the national level to support such workers which can be developed in related researches. Conclusions done in line 475 - 477 are unconvincing and they narrow the value of the research.

As suggested by the reviewer the conclusions of the study have been expanded (please see lines 529-557). We believe that the practical applications of the results are now included, which, in our opinion, increases the value of the study.

More comments see in the text of the article.

We are listing the rest of comments below, as well as our response to each of them.

These details can be omitted in the title.

As noted above, because another reviewer posed a similar request, these details have been omitted.

Abstract should not contain such general statements.

Because another reviewer posed a similar request, this general statement has been removed from the abstract

The approaches to form the samples should be described more detailed. Why did you choose these 4 groups? Which professional boards were involved to get the contacts?

Regarding these four groups, we think that we provided an explanation for the reason to be chosen as our sample in the Introduction (please, see lines 58 to 68). In that paragraph, after listing the essential workers considered by the Spanish government during the lockdown period, in which it is visible that health, information, food and protection were the main areas in which workers had to continue working, we argue that “in accordance with both the Spanish and the international criteria, in this study we considered frontline workers as healthcare workers, protective service workers, cashiers in grocery and general merchandise [11], as well as journalists working during the Spanish lockdown period”.

With regard to the professionals boards that we contacted, as noted above, because another reviewer posed a similar request, we have described more in depth the procedures section to address the reviewer’s comments (See lines 230-251).

Please, provide the full list of items of IES-R or define the pages of these two references clearly in your citations. These sources are unavailable for free reading, so readers cannot "guess" 22 items included in questionnaire. In the given form it is standard references to both chapters with full text, not references to certain pages with IES-R indicators

We thank the reviewer for this comment. We agree with the need for transparency and free access in science and we indeed strive for it.

The references that the reviewer mentions, #42 and #43, are references that give credit to the original work in English. Please note that since our study was carried out in Spain, we used the Spanish scale by Baguena et al. (2001) (reference #44), which is freely available at https://dialnet.unirioja.es/servlet/articulo?codigo=7061433. We have indicated so in the appropriate reference, adding the URL at which the article is available and the date on which we first accessed it (see lines 690-691).

If the reviewer wishes to access the IES-R scale in English language, the 22 items have been included in the second page of this recent and freely available article: https://hign.org/sites/default/files/2020-06/Try_This_General_Assessment_19.pdf (please note that it only reproduces the scale to facilitate professional use, it is not the original publication

College of Physicians- was the only board you contact with?

As stated above, we have provided more detailed information about the boards contacted to be more transparent (please, see lines 230-251).

Sencence - The years of experience was only significantly -and inversely- Unnecesary hypen?

We have removed the hyphens (see line 360)

Conclusions: Are these measures can be defined as "urgent measures are needed" as it is written in the end of the Abstract?

Following the reviewer’s suggestion, we have deleted the last sentence of the abstract. Indeed, the measures should not be urgent but they should entail systematic modifications that would surely take their time. Consequently, we have added the sentence: “We suggest some organizational measures for frontline institutions, such as periodical monitoring or inclusion of psychologists specialized in crisis-management to prevent negative symptoms and provide timely support”.

Acknowledgments: invaluable?

We have replaced unvaluable for invaluable. Thank you very much for noticing the mistake.

Reviewer 3 Report

It's a good research but needs to improve the manuscript. The title, introduction, and discussion are too long. In methods, section 2.2. instruments, please explain the instrument first, and then the use of it. For example: "demographics" is not an instrument. A lot of tables are attached.

Author Response

Reviewer 3:

It's a good research but needs to improve the manuscript. The title, introduction, and discussion are too long. In methods, section 2.2. instruments, please explain the instrument first, and then the use of it. For example: "demographics" is not an instrument. A lot of tables are attached.

We have shortened the title, now it reads “Working in the times of COVID-19. Psychological impact of the pandemic in frontline workers in Spain”.

The introduction and the discussion sections are slightly longer than two pages each. Thus, we agree with the reviewer that they are not short, however we truly consider that it is important to maintain their current extension so that all the necessary information to understand the article is included (e.g. previous literature about every professional group included, conclusions referring to every relevant result). Please, consider that our article includes four different samples and multiple variables. Taking this into account, we have done a great effort to present only the most relevant information.

In the instruments section, the instruments are first described and then the use of it. A minor change has been made in the IES-R description to fulfill this comment (see lines 189-190).  We also replaced the term “demographic” by “demographic questions”.

We agree the reviewer that a lot of tables are included along the manuscript. However, after reviewing each of them carefully we consider that none can be removed without missing essential information. Our article includes multiple variables in four different subsamples, and all the analyses are presented in the tables.

Round 2

Reviewer 2 Report

All previous comments are considered. Authors made appropriate changes. In the given form description of methodology and obtained results are clear and more scientifically sound. The title reflects the essence and aim of the research obviously and without unnecessary details. 

The manuscript is acceptable for publication.

Author Response

Many thanks for your positive feedback. We think that thanks to your constructive comments the manuscript is improved now.

Reviewer 3 Report

I believe that the authors should improve the synthesis of the information as I said before. The introduction section is very extensive. I think you have to know how to say things in a few words. We must think about facilitating the reading of the readers, and not pretend to put all the information. I understand that you have done a lot of work, but understand that you must transmit the information in a way that is accessible to people. Are all these tables really necessary? From my point of view, it is clearly not necessary. The work of synthesis is complicated and takes time, polishing the data, dedicating time, is a function of the researchers.

I think they have not understood the suggestion I made about the instruments. Since when are "demographic questions", "work-related information" and other instruments? An instrument is, for example, a test or a questionnaire. This section is poorly expressed. In any case, put measures in place, but you have to know how to call things by their name.

I don't understand why they put tables in the discussion. It doesn't make sense. Excuse me for saying so, but the discussion is to discuss results, not to show more data. Highlight the results of your work and discuss them with findings, reflect on future lines, mention limitations... There are so many aspects to be addressed in a discussion, and you put up a table of results? I can't understand it.

As for the references. The first reference translates into English what the government of Spain has done, and the rest is left in Spanish. Coherence, please. If it is a reference in Spanish, just put it in Spanish. Respect the official sources. The same happens with reference 3, 7, 8, 49. I consider reference 55 not adequate since there are better sources to talk about the lack of nursing personnel.

I am sorry to tell you that from my point of view, even though research is good, if it is not presented in an adequate manner it loses a lot of value, for this reason, I consider that if the authors are not capable of synthesizing the information, this article should not be published. We should give priority to the publication of publications of methodological quality (which is not very strong research at the methodological level either), together with the way of presenting the results to the population. I would not like them to take it personally, I only look for the quality of the publications to evolve but it cannot be allowed to publish something that from my point of view is not polished. There is a lot of work behind, I am not denying it, but to publish from my point of view, more effort is needed.

Author Response

We appreciate the feedback provided by the reviewers, and we are really glad to see that Reviewers 1 & 2 consider that we have made all the requested modifications on the manuscript and that they recommend the publication of the article in its current form. We would also like to acknowledge the feedback provided by Reviewer 3, as well as his/her recognition of the work behind our study. After considering each of the comments made by Reviewer 3, we provide our detailed responses next:

  • The reviewer suggests the need for synthesizing the information provided in the introduction. We have carefully read the introduction again, thinking about how to synthesize the information provided as much as possible. After doing so, we truly consider that the introduction section only includes the necessary information to understand the article. Next, we provide a detailed response to justify our decision:
    • We first provide a brief contextualization of the COVID-19 situation in Spain. Studies focused on a single nation-wide perspective must provide specific nation data so that we researchers do not commit the error to generalize data that might not be analogous to the situation of other countries. We think it is particularly necessary that this is briefly detailed in the introduction because Spain is facing a situation significantly different to that of other countries in Europe (being the most affected country until Belgium recently surpassed us in number of contagions). Moreover, detailing specific nation circumstances makes our study meaningful and also provides transparency and contextualization, which are two main principles of open science that we would like to comply with.
    • We secondly describe the psychological impact on the general population. As one of our research aims is to compare the psychological state of the “general population” with that of the “frontline workers”, we deem necessary to briefly describe what it is known about one of the groups of comparison.
    • Then, we describe which professions were considered frontline workers and which difficulties did face each of the four groups we included in the study. We judge this as necessary because not all the groups that comprise frontline workers were assessed (for example, we did not include firefighters or bank workers), therefore a justification for our election should be provided to the readers. The specific difficulties faced in Spain were indicated so the ulterior psychological impact could be deemed in its proper context (as other countries might have faced different contexts or might have assigned different roles to the same frontline workers we were assessing). Given that occupational health always takes the cultural scenario into account, this information is very relevant to the audience of the section of the journal for which this manuscript is addressed (“Occupational Health”). Finally, the two previous reviewers specifically requested  this to be added, so this way we fulfilled their pertinent requests.
    • Finally, the introduction finishes with a brief paragraph describing the purpose of the study. As this is the classical structure found on scientific literature, we find it quite inadequate to dispense this information.

After having provided a justification for each section in the introduction to be maintained, please note that the introduction is only two pages long out of a total of 27 pages. In the case that the reviewer is still convinced of the need to reduce information, we would be very grateful to hear  suggestions about which parts can be eliminated without reducing the transparency and compliance with previous requests we would like to maintain throughout the introduction.

  • Regarding the amount of tables, we do not consider that any of them can be removed. We understand that an article with so many results is complex but drawing conclusions from results that are not reported is unethical, since in science it seems fundamental to back each statement and conclusions with data. We have also thought about deleting the non-significant results as an available option for reducing the number of tables, however, cherry-picking is incompatible with transparency and non-significant results are also relevant data that can illuminate occupational researchers about what not address with their workers, thus saving precious time and resources in their departments. Still considering the reviewer’s point, we have also thought of the option of dividing all this information into two separate articles, but that would be unethical as we would incur in salami publishing. Thus, considering all this, we have arrived at the conclusion that we cannot reduce the number of Tables.
  • As for the suggestion made about the instruments, it is completely true that “demographic questions” and “work-related information” are not psychometrically validated instruments, but they were relevant variables that were measured (and also provided significant correlations with psychological impact). Thus, we think we should describe them not only for the sake of transparency but also to enable study replication, which is a main point of open science. Thus, trying to comply with the reviewer’s request and also with the open science principles, and in order to avoid the possible terminological confusion, we have now entitled the section as “Materials” instead of “Instruments”. Hopefully this modification makes the text more appropriate.
  • With regard to the tables in the discussion, we would like to clarify that, according to the journal formatting requirements and author guidelines that we need to follow for submission, “tables must be placed in the text after being named, as close as possible from the place where they were named”. As tables 6 and 7 were named in the results section, but in the very next page the result section finished and the discussion section started, they were forcedly placed the on next page (therefore, after the discussion section had started). We have now tried to avoid any possible confusion by placing the start of the discussion section right after Table 7. Hopefully, we are convinced that the editorial service may be able to help at this point providing a nice format if the study is accepted for publication. In case the reviewer was referring to the fact that we included “see Table XX” in the discussion a few times, we decided to do so just to make easier for the reader to know which specific result we were discussing. However, trying to also comply with the reviewer’s potential point, we have also eliminated the specific references to the tables in the discussion.
  • As for the references, we understand the reviewer’s point and have modified the mentioned references. We have now included the name of the publication in both languages: Spanish and English. We believe that it is essential to also include the English translation of the title of each reference to make the information accessible to non-Spanish-speaking readers. This is common practice in reference formatting such as APA style. In any case, we are open to modify the format of our references in order to fulfill the journal requirements.
  • Also, we are in agreement with the reviewer and have replaced reference 55 for a more adequate reference (we have now used data from the Global Health Observatory of the World Health Organization which shows that Spain is one of the European countries with the lowest number of nurses per 10,000 inhabitants).
  • We truly regret that the reviewer considered that the article does not have a high level of methodological quality. We have used the same instruments that other international researchers have used in their studied. For example, the IES-R has been used in mostly all the studies worldwide investigating psychological impact during Covid-19, as this meta-analysis shows (Ren et al., 2020; Psychiatric Quarterly), which includes a study published in the International Journal of Environmental Research and Public Health (Wang et al., 2020). With regard to our method (online questionnaire), it is common practice in this type of studies and, moreover, it was the only way to approach frontline workers during the lockdown period (see meta-analysis by Ren et al., 2020). Finally, with regard to our analysis, we have used the proper statistical methods to the kind of data we had and provided a detailed explanation of why we chose them in the text (e.g., specifying the kind of variables and within or between analysis, see lines 244-250).
